# Exploring the Implementation of Workplace-Focused Primary Prevention Efforts to Reduce Family Violence in a Regional City: The Need for Clarity, Capacity, and Communication

**DOI:** 10.3390/ijerph192416703

**Published:** 2022-12-13

**Authors:** Caroline Sarpy, Heidi Shukralla, Heath Greville, Sandra C. Thompson

**Affiliations:** 1School of Health, Georgetown University, Washington, DC 20057, USA; 2School of Allied Health, University of Western Australia, Perth, WA 6009, Australia

**Keywords:** primary prevention, family and domestic violence, social norms, occupational health, gender equality, violence against women, community attitudes

## Abstract

In response to the high burden of family and domestic violence (FDV), The Australian National Plan to End Violence Against Women and Children has established that primary prevention measures are necessary to reduce FDV’s harmful impacts on health. The Community, Respect, and Equality (CRE) project is a primary prevention initiative aimed towards changing harmful social norms and practices that enable FDV in Geraldton, Western Australia. Organizations affiliated with the CRE are required to promote gender equality and a respectful work environment. However, there is a gap in the literature regarding the impact and effectiveness of such interventions, especially in rural/regional areas. As such, this study served to evaluate the project’s effectiveness in a CRE-certified workspace, a local non-profit social services provider. Investigators conducted interviews to learn how the organization had implemented the CRE, and whether the CRE had had an impact on social norms and practices within the work environment. Findings indicated that the project had largely failed to permeate workplace culture due to a lack of effective promotion, low perceived benefits, and low resources. Future interventions must take persuasive measures, even for organizations perceived to be receptive to change.

## 1. Introduction

Family and Domestic Violence (FDV), defined here as the threat and/or realization of physical, sexual, and/or emotional abuse, is a significant public health issue in Australia; intrafamilial abuse is the leading risk factor for injury, disability, and mortality in women aged 18–44 [1,2]. FDV also places victims at greater risk for engagement in behaviors that can negatively impact health, such as alcohol abuse and tobacco consumption, due to concomitant conditions, such as depression, anxiety, and/or emotional distress [3]. The Australian National Plan to End Violence Against Women and Children (National Plan) has established that a comprehensive response to appropriately address the issue must include primary, secondary, and tertiary interventions [4]. Primary prevention interventions have the potential to reach the “towards zero” goal, or the elimination of FDV. This is possible as primary preventions are designed to stop violence before it starts through measures aimed at primary causes for all persons, rather than targeted groups. [3,4]. Such a goal necessitates the contextualization of FDV as an issue requiring an intersectional lens, as FDV is a public health issue driven by social structures, such as gender inequality, racism, colonialism, and homophobia, among others [4,5].

### 1.1. FDV Prevention Framework

In response to the need for a national, holistic primary prevention framework, the Australian government formed Our Watch, a non-profit organization designed to drive social change across all levels of society [4,6]. Our Watch identifies four key drivers of gender inequality: condoning of violence against women, men’s control of decision making and limits to women’s independence, gender stereotyping and dominant forms of masculinity, and male peer relations that augment outdated masculine stereotypes [7]. Because these social phenomena are enabled by interrelated levels of society, such as laws that discriminate against women at the institutional level, unequal pay conditions at the organizational level, and attitudes at the individual level, Our Watch outlines the need for an approach to primary intervention rooted in the socio-ecological model of health and wellness [7]. Such an approach can address all levels and, therefore, has the power to change social norms, or the perceived notion of what constitutes “normal” in relation to a given situation. This is vitally important in order to meet the National Plan’s goals; even if an individual recognizes that sexism and violence against an individual based on their gender is wrong, a widespread perception that such behaviors are “normal” may prevent action on that individual’s part [8]. It is for this reason that the socio-ecological model has been used in coordination with FDV prevention, as a prevention approach involves change on a scale beyond solely the individual to address harmful social norms that enable detrimental health behaviors [7]. A 2019 study conducted by Webster et al. found that an individual’s attitudes towards gender inequality can also vary along a spectrum including “public” and “private” spheres, adding evidence to the argument that interventions require an intersectional, socio-ecological perspective.

FDV has a large impact on the workplace due to its association with increased risk for legal liability, workplace violence, and employee absenteeism and/or turnover [9]. In contrast, workplaces that have supportive cultures and policies related to FDV have the power to reduce negative outcomes, such as termination, absenteeism, and/or discrimination in the workplace, which is crucial to enable financial independence and self-sufficiency for victims [2]. As such, the National Plan and Our Watch have prioritized creating effective preventative interventions in the workplace [4,7].

### 1.2. The City of Greater Geraldton and FDV

Within the City of Greater Geraldton, in the Midwest Region of Western Australia, the Community, Respect, and Equality (CRE) agreement was developed following community engagement, and it aligns with the national priorities established by Our Watch. [6]. From 2018–19, Geraldton experienced a familial assault rate 47% higher than that of regional WA, and nearly three times greater than the state average [1]. This higher incidence of FDV evidenced a need for a comprehensive FDV primary prevention intervention to reach beyond the individual and instead tackle deeper rooted issues through addressing social norms and practices. The CRE is currently coordinated by Desert Blue Connect (DBC), an organization that offers family violence, sexual assault, and crisis management services for the Midwest region [10]. Affiliated organizations were originally intended to be coordinated by a CRE Reference Group, but staff shortages have prevented the group from providing adequate guidance. As such, organizations keep in communication through monthly communication meetings, which act as a means of keeping participants up to date with the project [11]. This has been done in coordination with the Conversations for Change project created by the Western Australian Centre for Rural Health (WACRH), a program with the aim of supporting the CRE in creating community-based dialogues about FDV [6].

The CRE requires affiliated organizations and businesses to commit to preventing FDV by promoting gender equality, changing workplace culture, and changing social norms that drive FDV [6]. This is accomplished through conducting at least seven evidence-based interventions, such as bystander trainings, policy changes, and providing resources and education [6,7]. The CRE also benefits from its gender-transformative approach, meaning that its FDV interventions address the social aspects of the issue, rather than isolating FDV as a separate issue from gender inequality [6,12]. This approach has been shown to be more effective for generating social change in previous studies [13,14].

### 1.3. Case Study Justification

Primary prevention approaches involve long-term change, and require ongoing monitoring and evaluation [7]. As such, a need exists for evaluating the effectiveness of programs that utilize the Our Watch framework, especially in regional and remote regions, which often observe higher rates of FDV as compared to urban areas and may require specialized intervention [6,7,15]. Consequently, this study utilized a local social services provider as a case study to provide an in-depth analysis of the CRE’s impact in the workplace. The organization functions as a not-for-profit auspiced by the Catholic Church, operating within the Midwest and Gascoyne Regions of Western Australia. Services offered include family counseling, separation, and support services, as well as educational workshops on family violence prevention and maternal care. The organization signed on to the CRE in July 2019, and is currently working towards Level One accreditation, which signifies that the organization is completing significant work towards implementing the seven evidence-based interventions required by the CRE. This organization was used as a case study for several reasons. First, it is an instrumental case, as an organization centered around supporting community members experiencing FDV should serve as an exemplar for a workplace with healthy views of both gender equality and FDV [16]. Therefore, analysis of issues related to gender inequality and its impacts within the workplace—both present and resolved—can better illuminate areas that encourage recognition of effective intervention implementation for the workplaces that also have stakes in community development and well-being.

Secondly, investigators theorized that because this organization has a strong commitment to gender equality, employees would bring these values and perspectives to both FDV victims and perpetrators in counseling services provided. This implicates that the organization’s workplace norms are particularly relevant in relation to community attitudes towards sexism, as well as health issues related to acceptance of behaviors linked to gender inequality.

Finally, the Our Watch framework has prioritized both the creation and evaluation of interventions’ impacts in rural and regional organizations with regard to the prevention of FDV [7]. This is due to the lack of evidence-based primary prevention effectiveness in the extant literature, as well as the concern surrounding the higher rates of FDV and limited FDV resources available in rural and regional areas [7]. Therefore, this paper serves to answer the questions of how an organization has implemented the CRE, the extent to which the CRE’s core messages have addressed social norms that enable FDV, and whether the CRE’s implementation has changed social norms within a professional context in regional Australia.

## 2. Materials and Methods

### 2.1. Research Design

The evaluation conducted by WACRH used qualitative data by conducting one-on-one interviews in order to gain comprehensive perspectives of participants’ thoughts and opinions on the CRE’s implementation. Interview questions posed to study participants drew from a socio-ecological perspective, which is to say, a holistic approach towards social norms and practices that acknowledges the interactions and effects of plural life settings, i.e., individual, community, societal, etc. [17,18]. Semi-structured interviews were used to collect data, as this approach has been found to be more effective for sensitive information and/or topics likely not discussed on a daily basis [19]. Individual interview length varied from 25 min to an hour based on personal stories, but were generally around 30 min. Questions explored changes in social norms, organizational capacity, employee support, personal experiences with sexism in the workplace, overall workplace culture, and any possible future interventions employees believe would be effective (Appendix A). Interviews were conducted on-site over four weeks by investigators from WACRH. Data were transcribed using Otter.ai technology, a secure A.I. program that translates speech to text, and was then checked by the first author. The resulting text was coded using thematic synthesis in Dedoose software version 9.0.62 (created by SocioCultural Consultants LLC, Los Angeles, CA, USA), with each quote falling into a “descriptive” theme, which was then sorted into “analytical” themes [20]. Pertinent quotes were used to illustrate each analytical theme found in the results. Interviews were conducted until themes began to repeat themselves amongst participants, suggesting that saturation had been reached [21]. Despite a relatively smaller sample size, the largely female workforce and underlying Catholic ethos may have provided reasonable grounds for saturation.

To encourage recruitment, management within the organization identified potential participants from several different service areas. This enabled data collection using a purposive sampling method known as the Matrix method [17]. This method provides several advantages for the purposes of this study. First, it generates stakeholder ownership and commitment; this is key in maintaining a strong relationship with CRE-affiliated organizations, which are participating voluntarily [17]. Second, it gives structure in participant selection; supervisors had the local knowledge of potential participants who were willing to speak with investigators, as well as of the organizational structure, so that investigators could stratify data according to service area and collect a diverse set of perspectives. In this case, employees from Executive Management, Administration Services, and Operation Services had the opportunity to communicate their experiences with the CRE. Within these service areas, there was further diversity in participant perspectives, as some service areas operate in different buildings, and the teams have vastly different roles. For example, those who spoke to us in Operational Services have responsibilities varying from the management of employee privacy and well-being to community engagement and volunteer coordination. This in-depth approach allows for data triangulation in order to minimize selection bias according to service area, and to empower investigators to compare attitudes based on subgroups within the organization [22].

Data credibility and validity was advanced by member checking, wherein each participant was emailed a copy of the final transcript from their interviews and given the opportunity to ensure that their words were not misconstrued and that their full meaning had been expressed [23]. This process allowed participants to confirm or reject the validity of data collected to prevent phenomena such as confirmation bias [23]. The opportunity to validate themes was also carried out through the presentation of interview findings (i.e., analytical themes) during an all-staff meeting. This allowed for further validation of data, as well as the opportunity to collect perspectives from employees who were not interviewed, reducing both confirmation and selection bias [24,25]. 

### 2.2. Participants

Eight full-time employees were interviewed by the research team. While inclusion criteria were decided by supervisors, interviews were only conducted with employees who were not volunteers, all of whom were eighteen years or older. Supervisors had also been encouraged to nominate employees from several different areas of the organization. Nine employees were invited to be interviewed by the research team, with one invitee declining for unstated reasons. The organization currently has 42 employees included on the payroll. Of these employees, 37 identify as female and five identify as male. As such, while supervisors had been encouraged to pick a diverse group of participants, seven of the eight interviewees were women.

### 2.3. Ethical Considerations 

After being nominated by supervisors, employees provided consent to share their contact information and communicated directly with investigators to schedule interview times. Participants were informed through both a comprehensive Participant Information Form and a consent form that they would be recorded for the purposes of collecting data for the report. Participants were also told that they were able to decline or stop participation at any time without any ramifications for their employment in order to enforce ethical rigor and minimize respondent bias [25]. 

The evaluation performed was part of the “Conversations for Change” project, which was approved by the Human Ethics board in the Office of Research Enterprise at the University of Western Australia (2019/RA/4/20/4860). 

## 3. Results

Interviews produced a diverse set of perspectives; however, throughout the process, several elements were common across the interviews conducted. Data analysis and triangulation identified four recurring themes running throughout. These themes are illustrated in Table 1 below. 

### 3.1. Positive Workplace Culture

Every interview conducted yielded at least one comment about the benefits of working at the organization. This positive environment was attributed both to the value-driven work as well as individual employees: *“We expect all of our staff to be respectful and treat everyone equally… It’s a part of our ethos. Because it’s justice, compassion, excellence… that’s our mission.”* [Participant 2, female]. Each participant described the ways in which their fellow employees and managers added to the work environment, through means such as dedication to teamwork, caring, and granting opportunities to better the organization. One participant recounted that when she encountered an issue at work several years prior, *“God bless, from my experience, it was listened to. On some level. Whereas for some organizations, it would have been shut down”* [Participant 3, female]. 

The work environment was universally agreed to be more supportive of the opinions and perspectives of female employees than that of other businesses, particularly spaces perceived to be male-dominated, such as Bunnings Warehouse (a household hardware company) or Woolworths (a popular grocery store chain). Several participants averred that the positive work environment was due in part to the value-driven work, but also recognized that the staff is predominantly female. One participant concluded that: *“ …we deliver service in a caring, empathetic professional manner. I think that’s reflective of… The predominant gender that we have”* [Participant 4, female]. The reasoning behind the disproportionate gender distribution focused on two arguments. The first stemmed from the idea that men are opposed to lower-paying jobs, which are more common in non-profit organizations. As one participant stated:


*“I remember when I moved from a particular place to come back here as a manager, and one of the guys said, ‘So what are you earning?’ I told him, and he said ‘Flippin’ heck, I wouldn’t even get out of bed for that.’ And I was managing! And I thought, ‘Yeah, and that’s the difference.”*
[Participant 5, female]

The second argument asserted that men are less likely to seek employment in workspaces such as social service providers because they perceive community care work as *“soft”* [Participant 6, female]. However, despite these perceived obstacles, the majority of participants reported that they were enthusiastic about hiring more male employees in the near future, as men have additional capabilities to engage with community members that female employees do not due to gender roles. As one participant said, *“…a few of them [clients] are coming in, have VROs [Violence Restraining Orders] against them so he’s good to have as a male there”* [Participant 1, female]. 

Employees also spoke enthusiastically of male coworkers being able to offer a different perspective in the workplace. One participant summarized the impact of a male presence in the workplace, saying: *“…it’s just so gorgeous having these five blokes on board, because they see things differently. And you sit there at the meeting like, hold on, tell me that again. Because I’ve just heard something that made me wake up and think* [Participant 5, female].”

### 3.2. Failure of the CRE to Engage with the Majority of Employees

Despite the engaged attitudes of interviewees and their generally supportive attitudes towards gender equality in the work environment, the majority of interview participants had not engaged with the CRE. All were aware of the project, but to varying degrees, with some participants unable to correctly recount the acronym, and most unsure of the actual goals of the CRE. Several did not make the connection between the CRE and FDV. When recounting the purpose of a recent training, one participant stated: *“Well it’s probably not the community, respect, equality, it’s probably more… About domestic violence”* [Participant 5, female]. Only one participant was able to clearly delineate the CRE’s goals, theory, and why it is important in the workplace. CRE engagement did not vary along service areas, but those who had been employed for longer periods were more likely to have engaged with the CRE. All participants agreed that the lack of engagement was due at least in part to insufficient promotion on the part of the organization: *“I believe that we became a part of the CRE… in 2020… But I don’t ever remember there being a presentation to start. So to be quite honest, I don’t know a lot about it”* [Participant 4, female]. This quote highlights the lack of promotion, as Centacare joined the CRE in 2019. 

Attitudes towards the lack of engagement varied. One participant asserted that:


*“I hope you have lots to recommend because I think it’s really slack that it sits on our like (email signatures) …That’s the base. That’s what you see. So if you’re sending out an email like we should know, what does that mean, as an organization, like what are we representing?”*
[Participant 4, female]

Others were less concerned with the project’s outcome and more concerned with the logic behind the program, believing the CRE to be redundant for their workplace: *“In many ways, it’s preaching to the converted and in fact it would probably be insulting. My headspace… is… talk to Bunnings, go have a talk to Woolworths”* [Participant 3, female]. These opinions were contradicted by anecdotes from other employees, who recounted previous incidents of employees that had suffered from FDV, as well as sexist incidents with colleagues that made them uncomfortable in the workplace. 

This data indicate that not only are employees struggling with acquiring the means to learn about the CRE and how to incorporate it into their lives, but some are also wary of adopting the CRE in the context of their workplace. According to the Health Belief Model (HBM), a theoretical framework used to guide health behavior intervention implementation, this implies that there are three key areas that must be addressed in order to ensure the CRE’s success: cues to action, perceived benefits, and perceived susceptibility [26]. A cue to action is defined as an impetus to begin practicing a certain health behavior. In the case of the CRE, there have been a limited number of cues introduced, as evidenced by the employees who knew very little about the program. Those that have been presented are in the form of various media campaigns, as well as educational training sessions. However, the lack of engagement on the part of employees indicates that the cues have not been successful. This is exemplified by one participant’s opinion on promotional stickers for the CRE:


*“I don’t even know that the CRE logo mentioned anything about family and domestic violence. Does it?... I know it says community respect, equality, but I don’t know whether the actual message is there or not… (after being informed of the current message of “Violence Is Never Ok’’) it could be any kind of violence, because that’s where I think they lost the primary focus on the purpose, what the whole CRE is about.”*
[Participant 2, female]

This quote illustrates the need for both an increase and a clarification of how the theory behind the CRE connects to real-life practices, and why this intervention is relevant for the workplace. 

Furthermore, CRE intervention activities must communicate why it is important to adopt its recommended behavioral and attitudinal changes. Because some employees have low perceived benefits of the program and low perceived susceptibility to sexism and FDV, future educational materials should draw a clear picture of both the importance and efficacy of this program to reduce the risk and/or seriousness of FDV [26]. Because of the possibility of a causal relationship between HBM variables, more effective CRE promotion could simultaneously address both of these areas [27]. 

### 3.3. Lack of Resources for CRE Implementation

Multiple participants informed the research team that learning about the theory and core messages of the CRE *“…hasn’t come from management…”* [Participant 4, female], and this was negatively impacting the organization’s capacity to implement the CRE. These participants mentioned that this resource gap had been accentuated by the COVID-19 pandemic, during which several longtime employees had left, and turnover was further accentuated by the rapid loss of some new hires. Thus, while management’s involvement was generally considered to be positive: *“They’ve taken the time to actually be… involved in things like the CRE, and other networking opportunities and different programs and things… they’re not someone who does not want to know. Willing to learn”* [Participant 1, female], there was still a disconnect between the presence of the CRE and its dissemination. This is exemplified in a statement by another participant:


*“I don’t know that I have seen any… Real changes in workplace culture. That I know. It was something that a previous manager was really driving. And we were sort of maybe getting there, but it never eventuated into anything that I’m aware of. Certainly there haven’t been sort of more, you know, big discussions around it or things like that.”*
[Participant 7, female]

This lack of adequate human resources to implement the CRE has been coupled with a lack of time resources, further decreasing general understanding of the intervention; employees described themselves as too busy to individually seek out more on the project. Moving forward, one employee proposed concentrating human and time resources into a facilitator from an external organization who can act as a coordinator for the organization, so that employees and managers face less individual burden to educate themselves and others on the topic: 


*“…somebody, say, in my position who, who went to these meetings before, the amount of work—and as I told you, the programs that I oversee, is very, very time consuming—and something important like CRE, which then I do feel it needs to get going again, I think it’s lost its way, it hasn’t had a coordinator. And so each organization is trying to do their own thing, whereas I think it needs to be overseen and then implemented within the organization… The fact (is) that I’m way too busy.”*
[Participant 6, female]

### 3.4. Evidence for Need of the CRE

Aside from anecdotal evidence, such as past recollections of employees struggling with FDV and sexist comments made in the workplace, additional factors suggested that an intervention such as the CRE would be appropriate for the workplace. A recurring theme of the interviews involved a responsibility to the community, as the organization often handles family matters that are delicate in nature. As such, employees unanimously agreed that they *“…work with [a] lot of vulnerable clients”* [Participant 2, female], and that healthy attitudes of employees with respect to gender equality and FDV prevention were crucial in order to execute their duties to the best of their ability. This sentiment of dedication to their work was reflected in employees’ reported involvement in continuous education resources, including workshops and conferences. Despite evidence from this study that current attitudes within the organization are largely conducive to a healthy outlook towards gender equality, some incidents of targeted sexism were reported. One employee adroitly summarized the current situation: “*I don’t think there’s any organization that has it 100% nailed… We’re not naive enough to think that we’ve got this beautiful Eden of a workplace”* [Participant 8, male]. This reported need to continually engage with educational resources in combination with the duty to ensure the well-being of community members points to the relevance of a primary prevention project such as the CRE. 

Interestingly, even participants who felt that their workplace did not have a need for the CRE, as compared to male-dominated spaces, praised the organization’s emphasis on continual education with respect to client procedures: *“…we’re always a part of anything that’s going on and all our staff are really well trained… and have attended numerous workshops on different topics”* [Participant 6, female]. Thus, there is a critical disconnect between recognition that a healthy workplace environment is beneficial for community members and the idea that the organization must continue to do internal work to eliminate gender inequality as part of the primary prevention of FDV. This again signifies that some employees have not made the connection between a respectful workplace and the issues confronting the community, including clients who frequent their services. 

Finally, many employees described the environment in ways that indicated they viewed the organization as dynamic and responsive. One respondent stated that the organization is *“…a fertile soil, we’re always changing, never the same*” [Participant 5, female]. This dynamism, while praised for its ability to generate new ideas, also prompted several employees to attest to the utility of the CRE due to a fear of negative change in the ever-evolving workplace environment. Interviewees vocalized concerns about new employees and/or management misunderstanding or disrespecting the mission and values of the organization. One employee stated that after the interview, she was motivated to learn about the project partially for the sake of potential new employees:


*“For me, I’m gonna go back and have a look at (the CRE). And I always look at things, and think how somebody who probably doesn’t sit in my headspace can understand what that’s about. Yeah, yeah, with somebody on the street, or a new employee, ‘cause there are always new starts coming through all the time.”*
[Participant 1, female]

Another interviewee echoed this sentiment, saying that she felt a recent training related to the CRE was helpful because of the need to speak up and reinforce organizational values in the event of turnover: *“I feel like in workplaces, people change, managers change, different staff come and go. One day, you might have to say [something]”* [Participant 2, female]. Again, the CRE was espoused as suitable for the workplace. 

## 4. Discussion

The interviews illuminated several key findings on the implementation of the CRE, as well as its impact and perceived utility in the workplace. However, the findings must be seen in the broader context of recent events impacting the CRE implementation, as the COVID-19 pandemic and Cyclone Seroja brought about widespread disruption in Geraldton [28,29]. These events, especially the COVID-19 pandemic, diminished the organization’s capacity to promote the CRE. Moreover, working from home and social distancing reduced the organization of CRE events with employees. Employee turnover—widely observed throughout the pandemic—also likely reduced general knowledge of the project among staff members [29]. 

Most employees interviewed were able to identify gender inequality as the primary driver of FDV. This finding contrasts with that of a 2019 study conducted by Puccetti et al., which found that key CRE stakeholders in the Geraldton community identified alcohol and drug abuse as the primary driver of FDV. The 2019 Local Community Attitudes and Exposure to Violence Survey (LCAEVS) conducted in the City of Greater Geraldton also reported that community members were more likely to associate substance abuse as a cause of FDV than gendered drivers of violence, such as sexist jokes between male peers [30]. However, it must be noted that the organization participating in this research specializes in family services, including counseling and education, and employees are generally well-versed in FDV theory. Hence, the findings might not be generalizable to the broader Geraldton community, although they may be relevant to organizations that offer similar services. 

Within the organization, while participants were largely able to connect gender inequality to FDV based on their background knowledge, only a minority were able to do so in the context of the CRE. This was attributed to an insufficient number of interventions, and a loss of the connection between the CRE and FDV in CRE promotional materials, raising issues of the communication of CRE messages. Furthermore, some participants saw the project as being redundant in a workplace that is geared towards reducing FDV in the community, failing to see the evidence that promoting CRE values would be beneficial for the workplace. Yet, even workplaces that have generally healthy workplace norms must make continuous efforts to combat issues such as sexism. The reported sentiments of projects such as the CRE being redundant for the evaluated organization’s workplace reflects a disconnect between changing social norms and community attitudes on a larger scale, and work on the part of the individual in collegial situations. 

There are several implications in these findings. First, there is a need for clarity in the messaging about the relationship between gender equality and the CRE, and communication of this in efforts for wider dissemination of information about the CRE and its goals. Employees cannot change behaviors and attitudes if they do not have the tools and knowledge to do so. Even community-oriented service providers, perceived to be open to intervention by CRE implementers, may require specialized interventions, which require further study in the future. Once again returning to the HBM, individuals’ baseline inclinations to engage with a practice are based on several factors, including available cues to action, perceived benefits, perceived barriers, perceived susceptibility, perceived severity, and self-efficacy [26]. In addition, individuals are influenced by their habits, by what is perceived to be ‘normal’ by others, and by the underlying attitudes and beliefs that uphold resistance to change. With the context of the HBM and external influences on its components, such as perceived benefits, in mind, one of the failings of the CRE’s implementation appears to be that CRE resources were disseminated with the assumption that participating organizations’ employees had high perceived benefits and susceptibility, and therefore would not require persuasion in order to adopt and promote the CRE’s values. However, in the case of the organization considered here, this was not accurate for all employees; some are comfortable with their organization’s norms and have the perception that their workplace is not in need of intervention, as compared to male-dominated workspaces. 

CRE interventions must not only increase the quantity of interventions, but also incorporate components of persuasion and clarity, as outlined by the HBM. For instance, multiple interviewees asserted that they believed the CRE to be confusing, as they could not independently make a connection between the language used in CRE materials and FDV. Complexity, or perceived difficulty in understanding an innovation, was therefore high in this case [31]. This relates to the issue of the CRE not explicitly naming FDV within promotional materials. However, based on the interviews conducted, this appears to have led to misunderstanding the critical element of individuals taking responsibility to combat gendered drivers of FDV in their own life and that of those with whom they have contact. Clarification of the theory behind the project would, therefore, be relevant to both those who are not familiar with the CRE and those who are currently confused by its message and, thus, less inclined to engage with it and/or promote it to others. 

Another relevant component is compatibility, which involves demonstrating how an intervention is compatible with pre-existing norms for a given population [32]. The organization examined explicitly states that its mission is centered around improving quality of life for everyone, and its Catholic origins strengthen internal values, such as justice and equality. Furthermore, interviewees reported that they have observed a strong workplace culture of gender equality and general encouragement to combat sexism both in the community and in the workplace. However, some interviewees reported sexist behaviors experienced in the workplace. Emphasizing the links between the CRE and pre-existing positive workplace norms in the context of a need for improvement could garner motivation for employees to engage with and, hence, strengthen the efforts of the CRE. Appealing to Christian values could also serve to bridge the gap between theory and practice [33]. 

Observability is also important in this case, as active, visible promotion of FDV prevention within the workplace could be beneficial to educate community members who seek out the organization’s services. The self-reported commitment from employees to bettering the community indicates that this approach would be well-received by the organization. Previous studies have shown this approach can be effective in bolstering awareness not only for the well-being of an individual, but also that of the broader community struggling with a health issue [34]. This is particularly relevant to FDV, where relevant behaviors are strongly tied to social norms [5].

### 4.1. Recommendations

While the CRE holds promise as an FDV primary prevention project, several efforts must be made to boost its promotion if it is to create change within a workplace environment. As previously mentioned, promotion must be modified in both quantity and content. Awareness and persuasion should be key focus points of future CRE activities in order to engage employees who have low perceived benefits and/or knowledge of the project. This could be accomplished by increasing the amount of CRE promotion in the workplace, and by modifying the language in promotional materials to reduce confusion about the intent of the project. A 2017 study conducted by Rapport et al. found that to strengthen diffusion efforts, language must over- rather than under-explain to stakeholders; excluding keywords hampers health promotion efforts. As such, direct inclusion of phrases related to gender inequality would likely be helpful for the CRE. 

In the context of the specific organization referred to here, CRE activities could be created that appeal directly to pre-existing organizational values and attitudes in the workplace. Cognitive principles of learning emphasize that personal experiences should be emphasized to increase engagement in an intervention [33]. This is supported by findings from another Australian FDV primary prevention program, Freedom From Fear. The program employed numerous stakeholders across Western Australia and successfully engaged with participants through relatable and emotional advertisements, such as children traumatized by FDV [34]. As such, CRE activities should be tailored to the organization at hand. For example, peer mediators could lead seminars to educate fellow employees on the connections between the CRE and Christian values of contributing to an individual’s dignity and well-being [35]. Dedication to community well-being could also be capitalized upon by emphasizing the “ripple effect” of creating an observable intervention. The inclusion of promotional materials in public spaces, such as waiting rooms, has been shown to increase awareness and motivation to engage in health behaviors [36]; however, they are not enough in themselves and should reinforce other communication. This activity is one of several possible evidence-based interventions that CRE-affiliated organizations can implement to fulfill their CRE requirement [6].

In terms of long-term changes, the lack of structural capacity to implement the CRE needs to be considered. Employees reported a lack of resources to plan and execute CRE activities internally, suggesting that an external organization may be necessary to help drive support for the project’s goals. Currently, DBC acts as the parent entity for the CRE. While DBC has an employee position specifically designed to support CRE development, the position has had personnel turnover and periods of vacancy, as was the case when this research was undertaken. Incentives must be made to encourage potential applicants to fill this role and further the CRE’s goals. 

### 4.2. Study Limitations

While the matrix method used in this study has strengths in stakeholder engagement and data stratification, it also introduces the possibility of sample selection bias, as there is the potential for supervisors to select interview participants based on their belief that these participants will communicate what they believe researchers “want to hear”, rather than the true experiences of employees [37]. However, the strict anonymity of the interview process, as well as the selection of employees from different areas of the organization, reduces the risk of bias [17]. The study was also limited to eight participants, which can result in the skewing of information due to a relatively smaller sample size [38]. Furthermore, seven of these eight participants were female, indicating that there is a gender distribution deviation from the general population, which may limit generalizability. Therefore, caution should be taken when generalizing the results, given the small sample size. 

## 5. Conclusions

The study yielded several findings of note. While the organization’s employees were able to make the connection between gender inequality and FDV—a divergence from characteristics of the general population in Geraldton—there were still widespread gaps in the CRE’s implementation. These gaps were attributed to several factors, including insufficient promotional materials and a lack of understanding of individual responsibility to change social norms in collegial settings. This erroneous understanding was linked to perceptions that the organization did not have a need for an intervention such as the CRE because its organizational mission already aligned with the project’s objectives. 

The implications of this are that efforts must be made to increase the quantity of CRE-related activities to boost awareness and engagement with the project. This also raises the issue of a need for a defined internal structure for project implementation; employee turnover was reported to be highly associated with a lack of engagement. In addition, the current absence of persuasive, clear language in CRE promotional materials must be addressed and remedied in order to maximize the project’s effectiveness. While the organization’s employees espouse views supportive of gender equality, they still have a significant role to play in standing up for these values with their colleagues and those who engage in their services. This responsibility must be communicated to employees. The combination of high background knowledge with low perceived need is a unique circumstance, yet one that is likely applicable to several CRE community service providers. Thus, the need for tailored interventions is critical to maximize program effectiveness. This must be heeded by those designing future interventions similar to the CRE.

## Figures and Tables

**Table 1 ijerph-19-16703-t001:** Key themes identified.

Theme	Elaboration
Positive workplace culture	Participants reported that the organization has an overall respectful, productive workplace environment conducive to the organization’s mission and accomplishing meaningful work in the community. Participants did report a need for more employees that identify as male.
Failure of the CRE to engage with most employees	Most participants reported that they had not engaged with the CRE as the original program designers had planned. There was a lack of educational efforts with employees regarding what the CRE is, the theory behind it, and its importance, as well as low perceived benefits of engaging with the CRE.
Lack of resources to implement the CRE	Both human and time resources were described as in short supply with respect to the CRE. This was due in part to the large employee turnover for those whose roles were associated with the CRE.
Evidence for need of the CRE	While many participants’ accounts detailed a positive work culture, several circumstances pointed to the aptness for an intervention such as the CRE. These included a self-reported desire to continue education on FDV and improve the workplace, a feeling of responsibility towards community members, and a fear of potential change in workplace culture with new employees.

## Data Availability

Not applicable.

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
