# Peer review of "Exploring the Implementation of Workplace-Focused Primary Prevention Efforts to Reduce Family Violence in a Regional City: The Need for Clarity, Capacity, and Communication"

_ijerph, 2022, doi:10.3390/ijerph192416703_

Round 1

Reviewer 1 Report

Thank you very much for inviting me to review this article. The authors discussed the implementation of the CRE, and its impact and performed utility in the workplace by using qualitative research methods. I think the structure of this article is complete and well written, but there are still some places that need to be further clarified.

-In the section of introduction, I feel that the introduction section lacks sufficient theoretical foundation. The authors can try to establish a theoretical analysis framework closely associated to FDV Prevention to better theorize and rationalize the application of FDV Prevention. Although the author mentioned the social ecological model, there was no in-depth discussion.

-In the section of method, 8 employees was interviewed in this study. The authors need to explain whether the data collection from only 8 employees can provide sufficient information and meet the principle of sample saturation. In addition, 7 of the 8 employees are female, which may mean that the findings of the study have greater explanatory power for female employees. The authors should knowledge that the gender distribution deviation in the sample is a limitation of the study.

-In the results section, when quoting the interviewees' words, whether it is possible to use a different font, such as italic.

-In the discussion and conclusion part, the authors need to compare and discuss the findings of this article with previous studies, and further clarify the contribution of this study.

Reviewer 2 Report

The aim of this paper is to examine uptake of the CRE agreement in a workplace dedicated to supporting community members affected by FDV. Its main contribution is to show through qualitative evidence that uptake has been limited by the lack of adequate resources to engage employees of the workplace in learning about CRE, as well as limiting beliefs of the employees about the CRE agreement and its usefulness in their setting.

General Comments:

While the authors focus on limiting beliefs of employees as an outcome of their collective attitudes that their workplace has adequate FDV understanding and gender equity, other explanations are not adequately considered. For example, there is insufficient recognition by the authors that training about CRE (when it does happen) should be adapted to the context of the specific workplace for better adoption. This is in alignment with the first tenet of Bybee's 5E psychoeducational model for learning that requires that learners be engaged through inclusion of their own first-hand experience in the teaching/training efforts. In other words, learners are less persuaded to learn about something if it is not viewed as relevant to their own and/or collective experience. In this paper, the authors should specify that the content of future CRE training must be adapted to the context of the workplace, which some participants described as not 100% gender equal, to gain better uptake of framework.

Authors explain limiting beliefs through the lenses of HBM and Diffusion Innovation Theory. This is confusing. Instead authors should choose one or the other as the primary lens for analysis and incorporate the other for particular features of the first. This is attempted, but not fulfilled.

Specific Comments

The word "paper" on line 102 should be replaced with "study" for improved accuracy.

There are some overly long sentences that limit reader understanding. For example, there is a sentence that starts on line 54 and ends on line 59.
